# A CLOSER LOOK AT THE WORD ANALOGY PROBLEM

## ABSTRACT

Although word analogy problems have become a standard tool for evaluating word vectors, little is known about why word vectors are so good at solving these problems. In this paper, I attempt to further our understanding of the subject, by developing a simple, but highly accurate generative approach to solve the word analogy problem for the case when all terms involved in the problem are nouns. My approach solves the word analogy problem using a small fraction of the data that is typically used to train word vectors. My results demonstrate the ambiguities associated with learning the relationship between a word pair, and the role of the training dataset in determining the relationship which gets most highlighted. Furthermore, my results show that the ability of a model to accurately solve the word analogy problem may not be indicative of a model's ability to learn the relationship between a word pair the way a human does.

## 1 INTRODUCTION

Word vectors constructed using Word2vec (Mikolov et al. (2013a), Mikolov et al. (2013c)) and Glove (Pennington et al. (2014)) are central to the success of several state of the art models in natural language processing (Kim (2014), Le & Mikolov (2014), Mikolov et al. (2013b), Vinyals et al. (2015)). These vectors are low dimensional vector representations of words that accurately capture the semantic and syntactic information about the word in a document.

The ability of these vectors to encode language is best illustrated by their efficiency at solving word analogy problems. The problem involves predicting a word, *D*, which completes analogies of the form '*A:B :: C:D*'. For example, if the phrase is ''*King:Queen :: Man:D*', then the appropriate value of *D* is *Woman*. Word2vec solves these problems by observing that the word vectors for *A*, *B*, *C* and *D* satisfy the equation $Vec(D) \approx Vec(C) + Vec(B) - Vec(A)$ in several cases.

Although this equation accurately resolves the word analogy for a wide variety of semantic and syntactic problems, the precise dynamics underlying this equation are largely unknown. Part of the difficulty in understanding the dynamics is that word vectors are essentially 'black boxes' which lack interpretability. This difficulty has been overcome in large part due to the systematic analyses of Levy, Goldberg and colleagues, who have derived connections between word vectors and the more human-interpretable count based approach of representing words. They show that 1) there are mathematical equivalences between Word2vec and the count based approach (Levy & Goldberg (2014b)), 2) that the count based approach can produce results comparable to Word2vec on word analogy problems (Levy & Goldberg (2014a)) and more generally, 3) that the count based approach can perform as well as Word2vec on most NLP tasks when the hyper-parameters in the model are properly tuned (Levy et al. (2015). Their results (see section 9 in Levy & Goldberg (2014a)) demonstrate that $Vec(B) - Vec(A)$ is likely capturing the 'common information' between *A* and *B*, and this information is somehow being 'transferred' on to *C* to compute *D*.

Still the question remains, how is this transference process taking place? The answer would provide insight into the topology of word vectors and would help us to identify gaps in our understanding of word vectors. In this paper, I attempt to gain insights into the transference process by building a simple generative algorithm for solving semantic word analogy problems in the case where *A, B, C* and *D* are nouns. My algorithm works in two steps: In the first step, I compute a list of nouns that likely represent the information that is common to both *A* and *B*. In the second step, I impose the information about the nouns obtained in the first step on to *C* to compute *D*. Both steps of the algorithm work only on word counts; therefore, it is possible to precisely understand how and why *D* is generated in every word analogy question.

Despite the simplicity of my approach, the algorithm is able to produce results comparable to Word2vec on the semantic word analogy questions, even using a very small dataset. My study reveals insights into why word vectors solve certain classes of word analogy problems much better than others. I show that there is no universal interpretation of the information contained in $Vec(B) - Vec(A)$ because the 'common information' between $A$ and $B$ is strongly dependent on the training dataset. My results reveal that a machine may not be 'learning' the relationship between a pair of words the way a human does, even when it accurately solves an analogy problem.

## 2 PROBLEM SETUP

**Problem Statement.** In this paper, I analyze a variant of the semantic word analogy problem studied in Mikolov et al. (2013c). The problem can be stated as follows: given 3 nouns, $A$, $B$ and $C$, appearing in the text, $T$, find a fourth noun $D$ such that the semantic relationship ($R$) between $A$ and $B$ is the same as the semantic relationship between $C$ and $D$. Here, $R$ describes an '**is a**' relationship; for instance, if $A = Beijing$ and $B = China$, then $R$ is likely to be *capital* since *Beijing* **is a** *capital* of *China*. Typically, the relationship between $A$ and $B$ will not be unique; in the example above, we could also have said *Beijing* **is a** *city* in *China*, or *Beijing* **is a** center of *tourism* in *China*.

**The dataset**: For my analysis, the text, $T$, comprises the first billion characters from Wikipedia. This dataset contains less than 10 % of the information present in Wikipedia. The data can be downloaded from http://mattmahoney.net/dc/enwik9.zip and pre-processed using wikiextractor detailed in http://medialab.di.unipi.it/wiki/Wikipedia_Extractor. The raw data is divided into several unrelated chapters for e.g., there is a chapter on 'Geography of Angola', 'Mouthwash Antiseptic' etc. As part of the pre-processing I remove all those chapters containing less than 5 words.

**Analysis questions:** I test the efficacy of my proposed solution using a subset of the semantic word analogy problems compiled by Mikolov et al. (2013c) that is relevant to this study. The test set used here comprise 8,363 problems [1] divided into 4 categories: common capitals (e.g., *Athens:Greece::Oslo:Norway*), all capitals (e.g., *Vienna:Austria::Bangkok:Thailand*), currencies (e.g., *Argentina:peso::Hungary: forint*) and cities in states of the U.S ( e.g., *Dallas:Texas::Henderson:Nevada*). The original dataset comprises questions in a fifth category of gender inflections (e.g., *grandfather:grandmother::father:mother*) which is left out of the analysis because many of the problems involve pronouns (e.g., *his:her::man:woman*).

**Word2vec.** I compare the results produced by my method with those obtained using Word2vec. Word2vec derives low dimensional vector representations for words such that 'similar' words occupy 'similar' regions in a vector space. Given a sequence of words $w_1, w_2 \ldots w_T$, Word2vec maximizes the log probability

$$J = -\frac{1}{T} \sum_{t=1}^{t=T} \sum_{-w \leq j \leq w} \log p(w_{t+j}|w_t), \tag{1}$$

where $w$ is the window size, and $p(w_{t+j}|w_t)$ is a function of the word vectors for $w_{t+j}$ and $w_t$ respectively; for detailed insights into the functioning of Word2vec see Goldberg & Levy (2014) and Rong (2014). Given the word vectors $Vec(A)$, $Vec(B)$ and $Vec(C)$ for $A$, $B$ and $C$ respectively, Word2vec derives $D$ as the word whose word vector best satisfies the relationship

$$Vec(D) \approx Vec(B) - Vec(A) + Vec(C) \tag{2}$$

The critical implicit assumption made in Equation (1) is that the $w$ words surrounding $w_i$ on either side, namely $[w_{i-w}, w_{i-w+1}, \ldots w_{i-1}, w_{i+1} \ldots w_{i+w-1}, w_{i+w}]$ contains the most semantically and syntactically relevant words to $w_i$. In the next section, I will explain how a slight generalization of this result forms the basis for my algorithm.

---

[1] The problems are the first 8,367 lines in https://github.com/nicholas-leonard/word2vec/blob/master/questions-words.txt

## 3 ALGORITHMIC DETAILS

The goal is to solve the word analogy problem using a simple, generative window based approach. I begin my analysis by noting that all terms relevant in the analysis ($A$, $B$, $C$, $D$, and $R$) are nouns. Accordingly, I construct a new document, $T'$, comprising only the nouns appearing in $T$, stored in the order in which they appear[2]

For convenience of notation, I will assume that there are $H$ nouns in $T'$, and the $i^{th}$ noun appearing in $T'$ is represented as $T'_i$ (i.e., $T'[i] = T'_i$). Since the same noun likely appear multiple times in the text, I use the set $Q(X) = \{i | T'_i = X\}$ to indicate the locations of the noun, $X$ in $T'$.

The key idea in Word2vec is that the words surrounding a given word in the text contain rich semantic information about that word. More generally, we expect the nouns surrounding a given noun to contain rich semantic information about the noun. This implies that for certain values of $w$, the context of $T'_i$ defined as

$$F(T'_i, w) = [T'_{i-w}, T'_{i-w+1} \ldots T'_{i-2}, T'_{i-1}, T'_{i+1}, T'_{i+2} \ldots T'_{i+w-1}, T'_{i+w}]$$
$$= [y | y \in T' d(y, T'_i) \le w] \tag{3}$$

will contain nouns that are semantically related to $T'_i$, for all $0 \le i \le H$ . In equation (3), $d(y, T'_i)$ is a metric distance function describing the number of nouns separating $y$ and $T'_i$ in $T'$, and $w$ is the window size. Clearly, equation (3) is likely to hold for small values of $w$, and not likely to hold for very large values of $w$. Accordingly, I make the following assumption:

**Assumption 1.** *There exists a maximal window size, $w^* = 2s$, such that the nouns present in $F(T'_i, w^*)$ are semantically related to $T'_i$ for all $0 \le i \le H$.*

Assumption 1 states that $T'_i$ is semantically related to $T'_j$ if $d(T'_i, T'_j) \le w^*$, and not semantically related to $T'_j$ if $d(T'_i, T'_j) > w^*$.

The results thus far describe contexts around one noun. We are interested in nouns that are semantically related to 2 nouns. Therefore, I define

$$\tilde{F}(i, j, w) = F(T'_i, w) + F(T'_j, w) | i < j \le i + w$$
$$= [T'_{i-w}, T'_{i-w+1}, \ldots T'_{i-1}, T'_i, T'_{i+1} \ldots T'_{j-1}, T'_j, T'_{j+1} \ldots T'_{j+w}], \tag{4}$$

which describes the combined context of $F(T'_i, w)$ and $F(T'_j, w)$, when $F(T'_i, w)$ and $F(T'_j, w)$ overlap i.e., when $i < j \le i + w$. For any 2 nouns, $A$ and $B$, and the set, $S(A, B, w)$ defined as

$$S(A, B, w) = \left\{ \tilde{F}(i, j, w) | i < j \le i + w, (i, j) \in (Q(A) \times Q(B)) \cup (Q(B) \times Q(A)) \right\}, \tag{5}$$

we have the following result:

**Proposition 1.** *If assumption 1 holds true, then all the nouns present in $W \in S(A, B, s)$ will be semantically related to both A and B.*

*Proof.* For every noun $N \in W$ we have

$$d(N, B) \le d(A, B) + s \le 2s = w^*, \tag{6}$$

i.e., $N$ belongs to one of the contexts of $B$ and is therefore, semantically related to $B$. Similarly, it can be shown that $N$ is semantically related to $A$. $\qquad\square$

Proposition 1 describes the ideal scenario; realistically, we do not expect assumption 1 to hold exactly and therefore, we expect $W \in S(A, B, s)$ to contain a higher frequency of nouns that are relevant to both A and B, and a lower frequency of nouns that are relevant to only either $A$ or $B$. In particular, the higher the frequency of the noun appearing in the list

$$L_C = [Y | Y \in W, W \in S(A, B, s), Y \ne A, Y \ne B], \tag{7}$$

---

[2] This is trivially achieved using any standard Part of Speech tagger. My analysis uses the one provided by spaCy (https://github.com/explosion/spaCy)

Table 1: **Efficiency at solving the word analogy problem.** Comparing the efficacy of the current approach with that obtained using word vectors trained on 1) the same dataset and 2) on the entire Wikipedia 2014 corpus, which contains more than 10 times the data. Performance on the Wikipedia 2014 dataset are taken from Levy & Goldberg (2014a)

| Category | current approach | Word2vec (same dataset) | Word2vec (full wiki 2014) |
|---|---|---|---|
| common capitals | 81.59 % | 63.44 % | 90.51 % |
| all capitals | 78.3 % | 23.80 % | 77.61 % |
| currencies | 0.7 % | 6.91 % | 14.55 % |
| city in state | 59.3 % | 23.79 % | 56.95 % |

the more likely it describes the relationship between $A$ and $B$. Since the relationship between $A$ and $B$ need not be unique, I assume that the set, $\mathbf{N}_{AB}$, comprising the $k$ most frequent nouns in $L$ are equally likely candidates for the relationship between $A$ and $B$. Algorithm 1 shows how $\mathbf{N}_{AB}$ can be derived from $A$, $B$ and $k$, for a given value of $s$.

---

**Algorithm 1** Algorithm for finding Candidate $\mathbf{N}_{AB}$

---

1: **procedure** CANDIDATE_N_VALUES$(T', A, B, k)$
2:      Set $L_C$ to empty List
3:      **for every** $(i, j)$ in $(Q(A) \times Q(B)) \cup (Q(B) \times Q(A))$ **do**
4:          **if** $F(T_i', s)$ overlaps with $F(T_j', s)$ **and** $j > i$ **then**
5:             $\tilde{F}(i, j, s) = F(T_i', s) + F(T_j', s)$          ▷ Equation (4)
6:             $L_C \leftarrow L_C + [w$ **for** $w$ **in** $\tilde{F}(i, j, s)$ **if** $(w \neq A$ and $w \neq B)]$     ▷ Equation (7)
7:      **return** MostCommon$(L_C, k)$        ▷ $\mathbf{N}_{AB}$ contains the k most frequent nouns in $L_C$

---

Once $\mathbf{N}_{AB}$ is computed using algorithm 1, we can derive $D$ from $C$ as the most frequently appearing noun in the list

$$L_D = [Y | Y \in W, W \in S(C, X, s), X \in \mathbf{N}_{AB}, Y \neq C]; \tag{8}$$

details for the computation of $D$ are provided in algorithm 2.

---

**Algorithm 2** Algorithm for finding Candidate D values

---

1: **procedure** CANDIDATE_N_VALUES$(T', \mathbf{N}_{AB}, C)$
2:      Set $L_D$ to empty List
3:      **for every** $X$ in $\mathbf{N}_{AB}$ **do**
4:          **for every** $(i, j)$ in $(Q(C) \times Q(X)) \cup (Q(X) \times Q(C))$ **do**
5:             **if** $F(T_i', s)$ overlaps with $F(T_j', s)$ **and** $j > i$ **then**
6:                 $\tilde{F}(i, j, s) = F(T_i', s) + F(T_j', s)$        ▷ Equation (4)
7:                 $L_D \leftarrow L_D + [w$ **for** $w$ **in** $\tilde{F}(i, j, s)$ **if** $(w \neq C$ and $w \neq X)]$    ▷ Equation (8)
8:      **return** MostCommon$(L_D, 1)$        ▷ $D$ is the most frequent noun in $L_D$

---

Algorithms 1 and 2 described above have two hyper-parameters: $s$ and $k$. A grid search on the parameter values suggests that the improvement of the approach begins to saturate around $s = 10$ and $k = 20$; these are the parameter values used in the remainder of the analysis unless specified otherwise.

## 4 RESULTS

Table 1 shows that the approach described in this paper does better than Word2vec on 3 out of 4 categories when the word vectors are trained using the same dataset. The current approach beats the results obtained by Word2vec in 2 out of 4 categories even when the word vectors are trained on the

entire Wikipedia 2014 corpus, which has more than 10 times the amount of data used in the current analysis.

Algorithm 1 assumes that the $k$ nouns in $\mathbf{N}_{AB}$ are equally likely candidates for the relationship between $A$ and $B$. While this assumption is a good starting point, it's not precise enough. Table 2 shows that the more frequently co-occurring nouns with *A* and *B* capture more information about the relationship between *A* and *B* than the less frequently co-occurring nouns. This suggests that 1) word vectors are likely capturing information about the nouns co-occurring with $A$ and $B$, weighted by their frequency of occurrence with $A$ and $B$, and 2) that the most frequently co-occurring noun with $A$ and $B$ likely represents the Maximum Likelihood Estimate (MLE) of the relationship between $A$ and $B$. Table 3 shows the 5 most frequently observed MLE values for questions in each of the 4 categories, listed by their frequency of occurrence. Although most of the MLE values in the table make intuitive sense, some (particularly the MLEs for 'common capitals') do not; I will revisit this point later.

The remainder of my analysis proceeds in two parts: In the first step, I discuss some of the problems associated with estimating word vectors for infrequently occurring words in the dataset and in the second step, I describe problems one might encounter with word vectors corresponding to frequently occurring words in the dataset.

LOW FREQUENCY WORDS

There is substantial variation in the prediction ability between the categories; both the current approach and Word2vec perform worse predicting currencies than the other 3 categories. This is probably because currencies appear far less frequently in the training dataset as compared to nouns in the other categories as demonstrated in Table 4. The lack of training data likely results in poor estimates of the relationship between *A* and *B* and accordingly, poor estimates of *D*. Similar problems will likely by observed with word analogy problems involving words appearing less frequently than the currencies in the current dataset.

Increasing the size of the dataset will resolve some of these issues relating to data scarcity, but to what extent? Will word vectors corresponding to most of the words trained on a larger dataset be accurate? To answer this question, consider the word vectors obtained by training Word2vec on the entire Wikipedia corpus, which comprises approximately 1.5 billion tokens, of which approximately 5.8 million are unique. The most popular token is *the* which appears 86 million times. [3]. From Zipf's law, we expect the frequency of occurrence of a particular word in the text to be inversely proportional to its frequency rank. Therefore, a word with a frequency rank of 1 million will appear approximately 86 times. Since Word2vec is fitting a 200 dimensional vector corresponding to this word using data from 86 points, I expect the estimate of the vector to be unreliable. This suggests that word vectors corresponding to at least 80 % of the unique words trained on the entire Wikipedia corpus will be unreliable.

In general, any dataset will contain some percentage of low frequency words for which accurate word vectors cannot be adequately estimated. Care should be taken to filter these out as is done in Pennington et al. (2014).

---

[3] http://imonad.com/seo/wikipedia-word-frequency-list/

Table 2: **Frequency of co-occurence matters**. Accuracy of the current approach at solving the word analogy problem for different values of $k$. Increasing the value of $k$ produces diminishing returns beyond $k = 5$; the improvements on going from $k = 10$ to $k = 20$ are nominal.

| Category | $k = 1$ | $k = 5$ | $k = 10$ | $k = 20$ (full model) |
|---|---|---|---|---|
| common capitals | 28.12 % | 70.31 % | 80.99 % | 81.59 % |
| all capitals | 26.09 % | 61.32 % | 71.13 % | 78.31 % |
| currencies | 9.7% | 4.2% | 1.87% | 0.7% |
| city in state | 27.6 % | 46.99 % | 55.78 % | 59.3 % |

Table 3: **Category wise MLEs** The 5 most frequently appearing MLEs for the value *D* in each of the categories, listed by their frequency of occurrence. The MLEs for 'common capitals' demonstrates that a model may not be 'learning' the relationship between a pair of words the way a human does.

| Category | Frequently observed MLEs |
|---|---|
| common capitals | city, war, de, republic, county |
| all capitals | city, capital, war, university, population |
| currencies | currency, south, dollar, nagorno, exchange |
| city in state | city, county, university, state, dallas |

HIGH FREQUENCY WORDS

In the 'common capitals' category, the countries and capitals being considered appear with high frequencies (see table 4), and are therefore not plagued by the data scarcity issues described in the previous section. Furthermore, the algorithm described in this paper is able to solve the word analogy problem in this category with a high accuracy as shown in table 1. These facts together seem to imply that the approach is learning the relationship *capital* accurately from the data. However, as shown in table 3, *capital* is not even in the top 5 most likely candidates for the relationship between *A* and *B*. This suggests that a model may not be 'learning' the relationship between a pair of words the way a human does, even when it accurately solves the word analogy problem.

To further elaborate on this point, consider figure 2 of Mikolov et al. (2013c), wherein the authors demonstrate that the projection of the vector describing the relationship between a *country* and its *capital* has nearly the same orientation for ten (*country,capital*) pairs. The authors attribute this fixed orientation to the ability of word vectors to learn the 'is capital' relationship without any supervised effort. Table 5 indicates that word vectors might actually be learning the relationship from information about wars afflicting cities in different countries. [4] Since the attacks in a war are generally targeted at the capital city, word vectors appear to be learning this relationship simply as a result of correlations. These results show that the context that a model uses to infer the relationship between a pair words may not match the context that humans typically use to infer the relationship between the same pair of words. The ambiguity in context can lead to some paradoxical properties of word relationships (see table 6) such as:

- **Pluralization can change the relationship between words**. The relationship for the pair *Bear:Lion* derives from the fact that they are both animals, but the relationship for the pair *Bears:Lions* derives from the fact that they are both names of sports teams. A corollary of this result is that the results of word analogy questions need not be symmetric; for the word analogy question *Bears:Lions::Bear:D* the context is 'sports game', but for the word analogy question *Bear:Lion::Bears:D* the context is 'animal'.

---

[4]Most of the relationships seem to derive from World War II

Table 4: **Median counts** Median number of times *A* and *B* appear in the dataset for each of the categories. The median counts for the frequencies are far less than that for any other group.

| Category | Median value of A | Median value of B |
|---|---|---|
| common capitals | 827 (A = Capital City) | 5344 (B = Country) |
| all capitals | 146 (A = Capital City) | 753 (B = Country) |
| currencies | 2709 (A = Country) | 47 (B = Currency) |
| city in state | 549 (A = City) | 5310 (B = State) |

Table 5: **Derived relationship between countries and their capitals**. The five most commonly co-occurring nouns with countries and their capitals. The countries chosen in this list are the same as those considered in figure 2 of Mikolov et al. (2013c). The model appears to be learning information about the wars that affected the capital city of these countries.

| *A* | *B* | $\mathbf{N}_{AB}$ |
|---|---|---|
| China | Beijing | republic, people, government, **captial**, taiwan |
| Russia | Moscow | **city**, petersburg, **war**, time, st. |
| Japan | Tokyo | **war**, world, **city**, airport, university |
| Turkey | Ankara | **city**, republic, treaty, **captial**, **war** |
| Poland | Warsaw | **war**, **city**, army, capital, germany |
| Germany | Berlin | west, east, **war**, **city**, republic |
| Italy | Rome | **city**, pope, **war**, who, bc |
| Greece | Athens | **city**, bc, games, **war**, olympics |
| Spain | Madrid | de, franco, years, barcelona, **war** |
| Portugal | Lisbon | king, **city**, spain, capital, john |

- **There can be ambiguity in the relationship, even within the same context**. *Bears :Lions* could be referring to the famed rivalry between the Chicago Bears and the Detroit Lions in the National Football League[5]. *Bears:Lions* could also be referring to the merger of the Brisbane Bears and the Fitzroy Lions to form the Brisbane Lions in the Australian Football League[6]. Here, there is ambiguity in the relationship despite the fact that in both cases, the nouns, *Bears* and *Lions* are being used in the context of sports teams. The relationship which gets over-emphasized in the word vector is strictly a function of the data being used to train the word vectors.

- **Inferences drawn from word analogies may produce counter-intuitive results**. The relationships for the *Lions:Giants* and *Dolphins:Giants* pairs are deriving from the fact that the Lions, the Dolphins, and the Giants are teams in the National Football League. However, the relationship for the *Lions:Dolphins* pair is deriving from the fact that they are both animals.

## 5    WHAT IS WORD2VEC DOING?

The generative approach described above makes a critical assumption that is not required by Word2vec – that the answer to the word analogy problem being posed will always be a noun. Indeed, Word2vec produces high accuracies even on questions where the answers to the word analogy questions are not nouns, by 'learning' the part of speech required by the outcome.

I believe that this learning takes place because word vectors corresponding to words having the same Part of Speech (POS) tag lie in the same subspace of the word vector space. The word analogy problem relies on the fact that

$$Vec(B) - Vec(A) \approx Vec(D) - Vec(C). \tag{9}$$

---

[5]https://en.wikipedia.org/wiki/Bears%E2%80%93Lions_rivalry
[6]https://en.wikipedia.org/wiki/Brisbane_Lions

Table 6: **Word relationship ambiguities** Ambiguities associated with determining the relationship between a word pair.

| *A* | *B* | $\mathbf{N}_{AB}$ |
|---|---|---|
| bear | lion | animals, grizzly, food |
| bears | lions | week, game, brisbane |
| lions | dolphins | sea, whales, seals |
| dolphins | giants | bowl, miami, super |
| lions | giants | games, detroit, season |

Table 7: **Phrase information and common information** Relationship between *A* and *B* when considered as part of a phrase (algorithm 3), and as two separate words (algorithm 1). The cases where the information provided by the two algorithms disagree are marked in red.

| A | B | Phrase analysis (algorithm 3) | Common information (algorithm 1) |
|---|---|---|---|
| Larry | Page | google, sergey, university | wikipedia, google, university |
| Canada | Air | airport, airlines, airline | airlines, airport, service |
| Baltimore | Sun | u.s., negroponte, newspaper | times, morning, washington |
| Montreal | Canadiens | team, hockey, nhl | cup, team, nhl |
| Stephen | King | novel, story, series | henry, who, death |
| Pacific | Southwest | airline, service, us | population, races, density |

Therefore, if a POS subspace did exist, $Vec(D)$ would belong to $Span(Vec(A), Vec(B), Vec(C))$, and $D$ would be forced to have the same POS tag as either $A$ or $B$ or $C$. For the word analogy questions considered in the previous section, $A$, $B$ and $C$ are all nouns, and therefore I would expect $D$ to also be a noun, as is the case.

WHAT IS VEC(B) -VEC(A) CAPTURING?

The idea behind equation (9) is that $Vec(B) - Vec(A)$ is capturing information that is common to both A and B, and the information that is common to both *A* and *B* is also common to *C* and *D*. If a POS subspace does exist, then for any 2 nouns, *A* and *B*, the 'common information' (described by $Vec(B) - Vec(A)$) will lie in the noun subspace of the vector space, and can be expressed as a linear combination of the information in other nouns in the text; this is the possibly why algorithm 1 is able to express the common information between 2 nouns solely in terms of other nouns.

The question remains, what does 'common information' mean? Is *Canada Air* the common information between *Canada* and *Air*? Our intuitive guess would be no, as is the claim made in Mikolov et al. (2013c), since there is nothing 'airline' about *Canada*. To test their claim, I construct algorithm 3 to derive the most likely interpretation of a two-word phrase 'A B'. Algorithm 3 is a minor variant of algorithm 1, which finds all nouns co-occurring with *A* and *B* under the stricter constraint that *B* is always the noun succeeding *A*. If the claim in Mikolov et al. (2013c) is true, we would expect the information captured by algorithm 1 to be drastically different from that captured by algorithm 3.

---

**Algorithm 3** Nouns closest to two word phrase 'A B'

---

1: **procedure** CANDIDATE_N_VALUES($T'$, $A$, $B$)
2:     Set $L_C$ to empty List
3:     **for every** $i$ in $Q(A)$ **do**
4:         **if** $T'_{i+1} == B$ **then**                  ▷ Stricter conditions than algorithm 1
5:             $\tilde{F}(i, j, s) = F(T'_i, s) + F(T'_j, s)$              ▷ Equation (4)
6:             $L_C \leftarrow L_C + [w$ **for** $w$ **in** $\tilde{F}(i, j, s)$ **if** $(w \neq A$ and $w \neq B)]$      ▷ Equation (7)
7:     **return** MostCommon($L_C$, $k$)          ▷ $\mathbf{N}_{AB}$ contains the k most frequent nouns in $L_C$

---

This is not what happens. Table 7 shows that both, algorithm 1 and algorithm 3 capture information about *Canada Air* being an airline equally well. But, when considering the common information between the words *Pacific* and *Southwest* , algorithm 3 captures information about the airline whereas algorithm 1 captures information about the region. Similar contradictions are observed when considering names of famous people – for both algorithms, the common information between *Larry* and *Page* is Google. But, with the words *Stephen* and *King*, algorithm 3 captures information about the novelist while algorithm 1 captures information about history.

These results suggest that the notion of 'common information' is very subjective, and is strongly influenced by the data used to train the word vectors.

## 6 CONCLUSION

Although problems have become a mainstay in illustrating the efficacy of word vectors, little is known about the dynamics underlying the solution. In this paper, I attempt to improve our understanding by providing a simple generative approach to solve the problem for the case when *A*, *B*, *C* and *D* are nouns. My approach proceeds by first estimating the relationship between the (*A, B*) pair, and then transferring this relationship to *C* to compute *D*. My results demonstrates the high ambiguity associated with estimating the relationship between a word pair, and the role of the training dataset in determining the relationship which gets most represented. My analysis shows that even when the model predicts *D* accurately, it is difficult to infer the relationship the model learns about the (*A, B*) pair.

ACKNOWLEDGMENTS

I would like to thank Nima Reyhani and Boris Ginsburg for useful comments on earlier drafts, which greatly improved the quality of the analysis. Special thanks to Maria Ginsbourg for meticulously proofreading and editing the paper.

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
