# OpenReview forum: "A closer look at the word analogy problem"
_ICLR.cc/2018/Conference — Reject_

### Official Review · AnonReviewer3 · 2017-11-25

**Rating:** 2
**Confidence:** 5

**Review:**

This paper proposes a new method for solving the analogy task, which can potentially provide some insight as to why word2vec recovers word analogies.

In my view, there are three main issues with this paper: (1) the assumptions it makes about our understanding of the analogy phenomenon; (2) the authors' understanding of the proposed method, what it models, and its relation to prior art; (3) the very selective subset of analogies that the author used for evaluation.


ASSUMPTIONS
The author assumes that there the community does not understand why word embedding methods such as word2vec recover analogies. I believe that, in fact, we do have a good understanding of this phenomena. Levy & Goldberg [1] showed that optimizing x for
argmax(cos(x, A - B + C))
is equivalent to optimizing x for
argmax(cos(x, A) - cos(x, B) + cos(x, C))
which means that one can interpret this objective as searching for a word x that is similar to A, similar to C, and dis-similar to B. Linzen [2] cemented this explanation by removing the negative term (-cos(x, B)) and showing that for a wide variety of analogies, the method still works. Drozd et al [3] and Rogers et al [4] also argue that the original datasets used by Mikolov et al were too easy because they focused on encyclopedic facts, and expand these datasets to other non-encyclopedic relations, which are significantly more difficult to solve using simple vector arithmetic.

In other words, we know why word2vec is able to solve analogies via vector arithmetic: because many analogies (like those in Mikolov et al's original dataset) are "gameable", and can be solved by finding a term that is similar to A and similar to C at the same time. For example, if A="woman" and C="king", then x="queen" fits the bill.


METHOD
From what I can understand, the proposed method models the 3-way co-occurrence between A, B, and a context noun (let's call it R). Leveraging the distribution of (X, R, Y) for solving problems in lexical semantics has been studied quite a bit in the past, e.g. Latent Relational Analysis [5] and even Hearst patterns. I think the current description overlooks this major deviation from word2vec and other distributional methods, which only model the 2-way co-occurrence (X, R). This is a much more profound difference than just filtering non-nouns. I think the proposed method should be redescribed in these terms, and compared to other work that modeled 3-way co-occurrences.


EVALUATION DATA
The author evaluates their method on a subset of the original analogy task, which is very limited. I would like to see an evaluation on (A) the original two datasets of Mikolov et al (without non-nouns), and (B) the larger datasets provided by Drozd et al [3] and Rogers et al [4].

In addition, since I think the analogy phenomenon is well understood, I would like to see some demonstration that this method has added value beyond the analogy benchmark.


MISCELLANEOUS COMMENTS
* The author does not state the important fact that when searching for the closest x to A - B + C, the search omits A, B, and C. It is often the case that the result is A or C without this omission.
* The paper is partially de-anonymized (e.g. links and acknowledgements).
* One of the problems with modeling 3-way co-occurrences (as opposed to 2-way co-occurrences) is that they are much sparser. I think this is a more precise explanation for why the currency relation is particularly hard to capture with this method.

[1] http://www.aclweb.org/anthology/W14-1618
[2] http://anthology.aclweb.org/W16-2503
[3] http://aclweb.org/anthology/C/C16/C16-1332.pdf
[4] http://www.aclweb.org/anthology/S17-1017
[5] https://arxiv.org/pdf/cs/0508053.pdf

---

### Official Review · AnonReviewer1 · 2017-11-27
**This paper shows some interesting preliminary results, but requires substantial additional work to be acceptable for publication**

**Rating:** 3
**Confidence:** 4

**Review:**

This paper presents, and analyzes, a method for learning word relationships based on co-occurrence.  In the method, relationships between pairs of words (A, B) are represented by the terms that tend to occur around co-mentions of A and B in text.  The paper shows the start of some interesting ideas, but needs revisions and much more extensive experiments.

On the plus side, the method proposed here does perform relatively well (Table 1) and probably merits further investigation.  The experiments in Table 1 can only be considered preliminary, however.  They only evaluate over a small number of relationships (three) -- looking at 20 or so different relationships would greatly improve confidence in the conclusions.

Beyond Table 1 the paper makes a number of claims that are not supported or weakly supported (the paper uses only a handful of examples as evidence).  An attempt to explain what Word2Vec is doing should be made with careful experiments over many relations and hundreds of examples, whereas this paper presents only a handful of examples for most of its claims.  Further, whether the behavior of the proposed algorithm actually reflects what word2vec is doing is left as a significant open question.

I appreciate the clarity of Assumption 1 and Proposition 1, but ultimately this formalism is not used and because Assumption 1 about which nouns are "semantically related" to which other nouns attempts to trivialize a complex notion (semantics) and is clearly way too strong -- the paper would be better off without it.  Also Assumption 1 does not actually claim what the text says it claims (the text says words outside the window are *not* semantically related, but the assumption does not actually say this) and furthermore is soon discarded and only the frequency of noun occurrences around co-mentions is used.  I think the description of the algorithm could be retained without including Assumption 1.

minor:

References to numbered algorithms or assumptions should be capitalized in the text.

what the introduction means about the "dynamics" of the vector equation is a little unclear

A submission shouldn't have acknowledgments, and in particular with names that undermine anonymity

MLE has a particular technical meaning that is not utilized here, I would just refer to the most frequent words as "most related nouns" or similar

In Table 1, are the "same dataset" results with w2v for the nouns-only corpus, or with all the other words?

The argument made assuming a perfect Zipf distribution (with exponent equal to one) should be made with data.

will likely by observed -> will likely be observed

lions:dolphins probably ends up that way because of "sea lions"

Table 4 caption: frequencies -> currencies

Table 2 -- claim is that improvements from k=10 to k=20 are 'nominal' but they look non-negligible to me

I did not understand how POS lying in the same subspace means that Vec(D) has to be in the span of Vecs A-C.

---

### Official Review · AnonReviewer2 · 2017-11-28
**Some nice ideas but needs more justification, and simpler explanation**

**Rating:** 3
**Confidence:** 4

**Review:**

This paper proposes a method to solve the 'word analogy problem', which was proposed as a way of understanding and evaluating word embeddings by Mikolov et al. There are some nice analyses in the paper which, if better organised, could lead to an improved understanding of semantic word spaces in neural nets.

comments:

The word analogy task was developed as an interesting way to analyse and understand word embedding spaces, but motivation for learning word embeddings was as general-purpose representations for language processing tasks (as in collobert et al, 2011), not as a way of resolving analogy questions. The authors develop a specialist method for resolving analogies, and it works (mostly) better than using the additive geometry of word-embedding spaces. But I don't think that comparison is 'fair' - the analogy thing is just a side-effect of word embedding spaces.

Given that the authors focus on the word-analogy problem as an end in itself, I think there should be much more justification of why this is a useful problem to solve. Analogy seems to be fundamental to human cognition and reasoning, so maybe that is part of the reason, but it's not clear to the reader.

The algorithm seems to be simple and intuitive, but the presentation is overly formal and unclear. It would be much easier for the reader to simply put into plain terms what the algorithm does.

Using a POS-tagger to strip out nouns is a form of supervision (the pos-tagger was trained on labelled data) that word-embedding methods do not use, which should at least be acknowledged when making a comparison. Similarly, it is nice that the present method works on less data, but the beauty of word embeddings is that they can be trained on any text - i.e. data is not a problem, and 'work' for any word type. Stripping away everything but nouns clearly allows co-occurrence semantic patterns to emerge from less data, but at the cost of the supervision mentioned above. Moreover, I suspect that the use of wikipedia is important for the proposed algorithm, as the pertinent relations are often explicit in the first sentence of articles "Paris the largest city and capital of France...". Would the same method work on any text? I would expect this question to be explored, even if the answer is negative.

The goal of understanding word2vec and embedding spaces in general (section 5) is a really important one (as it can tell us a lot about how language and meaning is encoded in deep learning models in general), and I think that's one of the strongest aspects of this work. However, the conclusions from this section (and other related conclusions in other sections) are a little unclear to me. Perhaps that is because I don't quite get algorithm 3, which would be mitigated by an intuitive explanation to complement the pseudocode. I'm also confused by the assertion that Vec(A) - Vec(B) conveys the 'common information' in A and B. How can a non-symmetric operation convey 'common information'. Surely it conveys something about the relationship between A and B?

Minor point:
"may not the be indicative of the model's ability to learn the relationship between a word pair the way a human does" (Abstract)
- I'm not sure we know how humans learn the relationships between word pairs. Are you referring to formal semantic relations i.e. in taxonomies in WordNet? This sentence seems dangerous, and the claim about humans is not really treated in the article itself.

The a+cknowledgements compromise the anonymity of the authors.

---

### Decision · Program_Chairs · 2018-01-29
**ICLR 2018 Conference Acceptance Decision**

**Decision:**

Reject

**Comment:**

This paper does not meet the acceptance bar this year, and thus I must recommend it for rejection.